# Tryptophan Metabolism in Bipolar Disorder in a Longitudinal Setting

**DOI:** 10.3390/antiox10111795

**Published:** 2021-11-10

**Authors:** Frederike T. Fellendorf, Johanna M. Gostner, Melanie Lenger, Martina Platzer, Armin Birner, Alexander Maget, Robert Queissner, Adelina Tmava-Berisha, Cornelia A. Pater, Michaela Ratzenhofer, Jolana Wagner-Skacel, Susanne A. Bengesser, Nina Dalkner, Dietmar Fuchs, Eva Z. Reininghaus

**Affiliations:** 1Department of Psychiatry and Psychotherapeutic Medicine, Medical University Graz, 8036 Graz, Austria; frederike.fellendorf@medunigraz.at (F.T.F.); martina.platzer@medunigraz.at (M.P.); armin.birner@medunigraz.at (A.B.); alexander.maget@medunigraz.at (A.M.); robert.queissner@medunigraz.at (R.Q.); adelina.tmava@medunigraz.at (A.T.-B.); abigail_pater@yahoo.com (C.A.P.); Michaela.ratzenhofer@t-online.de (M.R.); susanne.bengesser@medunigraz.at (S.A.B.); nina.dalkner@medunigraz.at (N.D.); eva.reininghaus@medunigraz.at (E.Z.R.); 2Institute of Medical Biochemistry, Biocenter, Medical University of Innsbruck, 6020 Innsbruck, Austria; johanna.gostner@i-med.ac.at; 3Department of Medical Psychology and Psychotherapy, Medical University Graz, 8036 Graz, Austria; jolana.wagner-skacel@medunigraz.at; 4Institute of Biological Chemistry, Biocenter, Medical University of Innsbruck, 6020 Innsbruck, Austria; Dietmar.Fuchs@i-med.ac.at

**Keywords:** bipolar disorder, psychoneuroimmunology, kynurenine to tryptophan ratio, tryptophan metabolism, IDO-1

## Abstract

Immune-mediated inflammatory processes and oxidative stress are involved in the aetiopathogenesis of bipolar disorder (BD) and weight-associated comorbidities. Tryptophan breakdown via indoleamine 2,3-dioxygenase-1 (IDO-1) along the kynurenine axis concomitant with a pro-inflammatory state was found to be more active in BD, and associated with overweight/obesity. This study aimed to investigate tryptophan metabolism in BD compared to controls (C), stratified by weight classes, in a longitudinal setting, dependent on the incidence of BD episodes. Peripheral tryptophan, kynurenine, and neopterin were assessed in the serum of 226 BD individuals and 142 C. Three samples in a longitudinal assessment were used for 75 BD individuals. Results showed a higher kynurenine/tryptophan in both BD compared to C and overweight compared to normal weight persons. Levels remained stable over time. In the longitudinal course, no differences were found between individuals who were constantly euthymic or not, or who had an illness episode or had none. Findings indicate that tryptophan, kynurenine, and IDO-1 activity may play a role in pathophysiology in BD but are not necessarily associated with clinical manifestations. Accelerated tryptophan breakdown along the kynurenine axis may be facilitated by being overweight. This may increase the risk of accumulation of neurotoxic metabolites, impacting BD symptomatology, cognition, and somatic comorbidities.

## 1. Introduction

Bipolar disorder (BD) is a chronic and lifelong psychiatric illness characterized by recurrent depressive and manic or hypomanic episodes. Historically, diagnosis and treatment considerations are solely based on clinical phenomenology while objective biomarkers are insignificant. Identifying biomarkers that provide evidence about the current stage of illness and predict the further course of the disease is an essential endeavor in BD research. Polygenic predisposition, neurotransmitter imbalances, epigenetics, chronic and acute psychosocial stressors as well as oxidative stress, inflammatory and immune processes are vital in BD’s aetiopathogenesis [1]. Chronic low-grade inflammation in acute illness episodes but also in euthymia is discussed as one of BD’s most important pathophysiological underpinnings [2,3]. Pro-inflammatory cytokines such as interleukin (IL)-6, IL-1ß, tumor necrosis factor (TNF)-α, interferon (IFN)-γ, and acute-phase proteins such as C-reactive protein (CRP) were found to be elevated in BD [4,5]. A meta-analysis revealed that many oxidative stress markers such as lipid peroxidation, DNA/RNA damage, and nitric oxide are significantly increased in BD compared to the general population, thereby supporting the impact of oxidative stress [6]. In the current study, we focused on immunobiochemical pathways of the cellular immune response that are mainly activated by IFN-γ, namely tryptophan catabolism along the kynurenine axis and neopterin formation. IFN-γ signaling also stimulates the formation of reactive oxygen species (ROS), and reactive nitrogen species (RNS) arise in the presence of nitric oxide. Oxidative triggers shape the immune response by stimulating and maintaining a cytocidal and defense milieu. However, they are also involved in the inflammation termination and initiation of restorative processes, pointing towards their importance in responding to challenges and subsequent feedback [7].

BD is accompanied by a higher prevalence of overweight and obesity than the general population [8]. This increased prevalence leads to a high risk for comorbid metabolic and cardiovascular diseases resulting in significantly lower life expectancy [9]. Additionally, obesity is associated with cognitive deficits [10], a higher number of affective episodes and suicide attempts, and shorter periods of euthymia [11]. Reasons for the higher prevalence rates are multifactorial, including psychopharmacological treatment [12], lifestyle and behavior [13], and, again, chronic low-grade inflammatory processes [14].

Tryptophan is an essential amino acid relevant to protein synthesis and presumably plays an essential role in the biological underpinnings of BD through its catabolic pathways [15]. Tryptophan can be converted to neurotransmitters 5-hydroxytryptamine (5-HT, serotonin), N-acetylserotonin, and further to melatonin. Another metabolic route is the kynurenine-axis which depends on the enzymes indoleamine 2,3-dioxygenases (IDO)-1 and 2 and tryptophan 2,3-dioxygenase (TDO) [16]. The activity of TDO, which is predominantly expressed in the liver, is substrate-regulated and thus relatively stable (at least in the absence of glucocorticoids). In contrast, the activity of IDO-1 is strongly induced in monocyte-derived cells by pro-inflammatory cytokines, mainly IFN-γ, but also IL-2 and -6 or TNF-α play a role, and is inhibited by anti-inflammatory cytokines such as IL-4 [17,18,19]. The serum/plasma kynurenine to tryptophan ratio has been established as a proxy for IDO-1 activity when paralleled by elevated levels of pro-inflammatory mediators such as neopterin [20]. Kynurenine is metabolized further into several neuroactive compounds; of most significance are kynurenine acid, 3-hydroxykynurenine, and quinolinic acid, which may impact neuropsychiatric disorders such as BD [21]. Significantly, psychosocial stress does also affect tryptophan metabolism [22,23,24].

Increased kynurenine concentrations and a higher kynurenine/tryptophan were found in overweight and obese humans compared to normal weight individuals [25]. Kynurenine is associated with inflammation processes as well as aging [26]. Additionally, sex-specific differences have been reported for increased concentrations of kynurenine mainly in male individuals with affective disorders over a six-week psychiatric rehabilitation program [27]. Psychiatry research is particularly interested in the pathways of monoamine synthesis interplay with the inflammatory homeostasis catabolism and their link to neuropsychiatric disorders. Increased activation of IDO-1 was shown to correlate with cognitive deficits in animal models [28] and humans with Alzheimer’s disease [29]. Besides other neurodegenerative diseases such as Huntington’s and Parkinson’s disease and several cancers and autoimmune disorders, the kynurenine pathway was investigated in neuropsychiatric disorders such as schizophrenia and BD in cross-sectional designs [30].

Recent meta-analyses by Bartoli and colleagues [31] and Marx and colleagues [32] found lower peripheral levels of tryptophan and kynurenine and further products in BD but an increased kynurenine/tryptophan reflecting IDO-1 activity compared to individuals without a mental disorder. Additionally, kynurenine/tryptophan in cerebrospinal fluid (CSF) was found increased in BD [33]. The same study group found increased kynurenic acid only in CSF and not in peripheral plasma in individuals with BD and previous psychosis compared to controls, while the CSF kynurenic acid concentrations were unchanged in BD patients with ongoing depression [34]. This finding implies that the metabolic crosstalk is very complex, and the role of tryptophan metabolism in BD likely has complex influences. Moreover, the availability of CSF is usually limited.

Human monocytes/macrophages and dendritic cells release the pteridine neopterin, mainly stimulated by IFN-γ [35]. During activation of the cellular immune system, increased neopterin levels can be found in the serum of patients [36]. A correlation with other immunobiochemical pathways that share upstream regulators, such as increased tryptophan catabolism along the kynurenine axis, is frequently observed. Elevated kynurenine to tryptophan ratio in relation to increased neopterin can be found frequently in conditions where the immune system is chronically activated, including infections, malignancies, cardiovascular and neurodegenerative disorders [37]. In the late 80s, the correlation of decreased serum tryptophan and increased neopterin with neurologic and psychiatric symptoms was reported in patients with HIV-1 infection [20]. Moreover, these immunobiochemical pathways are relevant for neuropsychiatric symptoms associated with chronic low-grade inflammation as e.g., in aging [38].

The results of studies investigating individuals with affective disorders are inconsistent. Hoekstra et al. [39] found lower neopterin levels in individuals with BD than those with depression and without any mental disorder independent of their symptomatic status. Van den Ameele et al. [40] did not find any differences between individuals with BD in different illness states compared to controls. In the preliminary results of our cohort [41], kynurenine concentrations and kynurenine/tryptophan were increased while neopterin levels of individuals with BD in an euthymic state were decreased to healthy controls. When comparing overweight BD patients with normal weight, BD kynurenine and kynurenine/tryptophan were higher, and neopterin levels were increased; too. In addition, the stage of illness was positively correlated with neopterin levels in patients with BD.

The current study had two main objectives. The first aim was to extend the pilot study results with a slightly overlapping study cohort [41], comparing a larger number of individuals with BD and controls taking weight differences into account. Furthermore, to our knowledge, historically, tryptophan, kynurenine, and kynurenine/tryptophan in BD have only been investigated in cross-sectional designs. Thus, the second aim was to analyze these parameters in a longitudinal setting in BD patients dependent on the incidence of illness episodes and euthymic states.

## 2. Materials and Methods

### 2.1. Participants and Procedure

The investigation was conducted at the Medical University of Graz, Austria, Department of Psychiatry and Psychotherapeutic Medicine as a part of the ongoing multi-center BIPLONG study. The BIPLONG study explores the relationship between BD and obesity, metabolism, lifestyle and cognitive function in a longitudinal setting. Individuals with BD-diagnosed with the Structured Clinical Interview according to the Diagnostic and Statistical Manual of Mental Disorders (DSM)-IV (SCID-I [42])—and individuals without a mental disorder (C) were included in the study. For a follow-up period of seven years, study participants were assessed bi-annually (BD patients) or annually (C). The assessment included sociodemographic parameters, illness and medication history, current psychiatric symptomatology measured with validated rating scales, fasting blood, cognitive testing, anthropometric measurement including body mass index (BMI), electroencephalography, cranial magnetic resonance imaging, stool analysis, and lifestyle questionnaires. Fasting blood samples were collected between 8.00 a.m. and 9.00 a.m., and then stored at −80 °C until thawed for the biological assays.

The local ethics committee (Medical University of Graz, Graz, Austria; EK-number: 25-335 ex 12/13) has approved the study in compliance with the current revision of the Declaration of Helsinki, ICH guideline for Good Clinical Practice and current regulations and registered at clinicaltrials.gov, accessed on 15 October 2021 (NCT04972708).

The current investigation included 226 individuals with BD and 142 C for comparing data at the first measurement time. Additionally, longitudinal data over three measurement time points (t1, t2, t3) were available of 75 individuals with BD. There was not enough comparable longitudinal data of C due to the clinical nature of the study. Individuals with BD were either inpatients or outpatients of the dedicated center for BD and, similar to C, had to be of legal age and provided written informed consent before they participated in this study. Exclusion criteria included the presence of chronic obstructive pulmonary disease, rheumatoid arthritis, systemic lupus erythematosus, inflammatory bowel disease, neurodegenerative and neuroinflammatory disorders (i.e., Alzheimer’s, Huntington’s and Parkinson’s disorder, and Multiple Sclerosis), haemodialysis and interferon-α-based immunotherapy. Further exclusion criteria for controls were the presence of lifetime psychiatric diagnoses (verified by SCID-I) and a first-grade relationship to individuals with psychiatric disorders.

### 2.2. Laboratory Methods

Analysis of tryptophan and kynurenine in serum was performed by reverse-phase high-performance liquid chromatography (HPLC) with fluorescence and UV detection as reported previously [43,44]. Samples were briefly treated with trichloroacetic acid to precipitate protein. As an internal standard, 3-nitro-L-tyrosine was used. Tryptophan was detected by its native fluorescence (excitation wavelength of 286 nm, emission wavelength of 366 nm), kynurenine, and 3-nitro-L-tyrosine at a wavelength of 360 nm. The limits of detection were 0.1 µmol/L for tryptophan and 0.5 µmol/L for kynurenine. Further information on method validation was reported previously [45,46]. Optimal chromatographic separation was achieved on a LiChroCART RP-18 endcapped column (55–4.3 µm, Merck, Germany) by using 15 mmol/L acetic acid-sodium acetate solution (pH = 4.0) as mobile phase. Validation parameters were linearity (R^2^ ≥ 0.998 for both tryptophan and kynurenine), interday precision (≤1.8% for tryptophan and ≤4.2% for kynurenine) and recovery (99.3% ± 0.4% (mean SD) for tryptophan and 102.3% ± 1.0% for kynurenine) [45].

The kynurenine to tryptophan ratio can be applied to indicate IDO-1 activity if paralleled by increased concentrations of pro-inflammatory mediators such as neopterin [20]. Neopterin concentrations were determined by an enzyme-linked immunosorbent assay (ELISA) (BRAHMS, Henningsdorf, Germany) with a sensitivity of 2 nmol/L neopterin.

### 2.3. Materials

The Hamilton Rating Scale for Depression (HAMD; [47]) is an external rating scale used to determine the severity of depressive symptoms in patients already diagnosed with depression. The items are rated with zero to two or four points. In the current version, up to 65 points can be awarded. A cut-off value was not planned initially, and therefore various values have been used to classify euthymia and subsyndromal groups over the years. The items 4–6 (each 0–2 points) measure insomnia, particularly difficulties falling asleep, being disturbed during the night, and waking in the early morning hours. The Young Mania Rating Scale (YMRS [48]) is an external rating scale to determine the severity of manic symptoms in patients with BD. The YMRS consists of eleven items with five levels of severity. Euthymia was defined as a HAMD score <14 and a YMRS score <9 points.

In addition, illness episodes since the last testing, including data about the type (depression/mania/mixed episode), duration, and treatment, were assessed by standardized questions and medical record review. With that method, the variable “no episode” was formed, including individuals with no episode, and “low illness load” was formed, including at least one (hypo)manic episode or only depressive episodes from t1 until t3. If there were episodes of different polarities or mixed features, they were categorized into “at least one manic episode”, as manias are known to have a more significant pathological impact on neurobiological processes [49].

### 2.4. Statistical Analyses

All analyses were performed with the IBM Statistical Package for Social Sciences (SPSS), version 25.0. Analyses of variance (ANOVAs; metric data) and chi-square tests (nominal data) were conducted to test for differences in descriptive variables (age, BMI, sex) between the group (BD versus C) and weight group (normal weight: BMI < 25 kg/m^2^ versus overweight: BMI ≥ 25 kg/m^2^). To calculate the differences in concentrations of tryptophan, kynurenine, the kynurenine to tryptophan ratio and neopterin between the BD and C group as well as between the normal weight and overweight group multivariate co-variance analysis (MANCOVA) with age and sex as covariates were used. For the individuals with BD at t1 only, partial correlations with tryptophan, kynurenine, ratio, neopterin and the HAMD items 4–6 with the covariates age, sex, BMI were conducted. Repeated measures ANCOVAs for the independent variables tryptophan, kynurenine, ratio and neopterin at three different time points were conducted for the factors (a.) euthymic over all time points versus not and (b.) illness episode between t1 and t3 versus no episode. The duration between t1 and t3, age (mean of all test points), sex and BMI (mean of all test points) were used as covariates. Error probabilities below 0.05 were accepted.

## 3. Results

### 3.1. Differences between Groups and Weight Classes

Table 1 shows demographic data and concentrations of tryptophan, kynurenine, neopterin, and kynurenine/tryptophan of individuals with BD and C sub-grouped in normal and overweight participants. Individuals with BD differed significantly in age, BMI, and sex proportion (*χ*^2^ = 7.703, *p* = 0.006) from C. Individuals with BD had significantly lower serum levels of tryptophan and higher kynurenine/tryptophan than C (independent of BMI and in the MANCOVA, see Table 1). Independent of BD and C group, participants with overweight in comparison to normal weight had significant higher levels of serum kynurenine (overweight 2.34 ± 0.83 µmol/L versus normal weight 1.96 ± 0.54 µmol/L; *F1,364* = 10.20, *p* = 0.002, *η*^2^ = 0.03) and significant higher kynurenine/tryptophan (overweight 40.35 ± 13.73 versus normal weight 33.73 ± 10.29; *F1,364* = 9.23, *p* = 0.003, *η*^2^ = 0.03). In the MANCOVA, using both group and weight as between subject factors, we found a significant difference in kynurenine with higher levels of overweight individuals than normal weights (see Table 1 and Figure 1).

There were no differences in neopterin levels either between the groups nor between the weight classes. Nonetheless, neopterin concentrations correlated in both groups negatively with tryptophan (BD: *r* = −0.179, *p* = 0.045; C: *r* = −0.271, *p* = 0.008) and positively with kynurenine (BD: *r* = 0.337, *p* < 0.001; C: *r* = 0.504, *p* < 0.001 and kynurenine/tryptophan (BD: *r* = 0.543, *p* < 0.000; C: *r* = 0.658, *p* < 0.001).

Of 204 individuals with BD at t1, the mean for falling asleep problems was 0.45 (±0.68), for sleep disruptions during night 0.41 (±0.66), and 0.30 (±0.63) for morning hours waking. No correlations between tryptophan or neopterin with the insomnia items of the HAMD were found. There was a significant negative correlation between the HAMD item 4 for difficulties falling asleep and kynurenine (*r* = −0.150, *p* = 0.033) as well as kynurenine/tryptophan (*r* = −0.170, *p* = 0.016). The other items 5 and 6 did not correlate significantly.

### 3.2. Longitudinal Setting

Table 2 shows the demographic, anthropometric and illness specific (euthymic at testing and illness episode in between test points) data as well as tryptophan and kynurenine concentrations and the ratio (*n* = 75) and neopterin (*n* = 68) of individuals with BD at three measurement time points. There was no significant difference in the chronological sequence in tryptophan (*F2,73* = 0.97, *p* = 0.385, *η*^2^ = 0.03), kynurenine (*F2,73* = 0.81, *p* = 0.450, *η*^2^ = 0.02), kynurenine/tryptophan (*F2,73* = 0.20, *p* = 0.820, *η*^2^ = 0.01) and neopterin (*F2,66* = 0.42, *p* = 0.658, *η*^2^ = 0.01).

No differences in tryptophan, kynurenine, kynurenine/tryptophan or neopterin were found when using the euthymic state at all three time points (no/yes) as inner subject variable (tryptophan: *F2,66* = 0.82, *p* = 0.447, *η*^2^ = 0.02; kynurenine: *F2,66* = 0.44, *p* = *0*.649, *η* = 0.01; kynurenine/tryptophan: *F2,66* = 0.14, *p* = *0*.870, *η*^2^ = 0.01; neopterin: *F2,59* = 0.14, *p* = 0.328, *η*^2^ = 0.04). Additionally, no differences in the targeted variables were shown between individuals with BD who had no illness episode or at least one episode with manic symptoms or one or more depressive episodes (tryptophan: *F4,134* = 0.81, *p* = 0.520, *η*^2^ = 0.02; kynurenine: *F4,134* = 0.37, *p* = 0.827, *η*^2^ = 0.01; kynurenine/tryptophan: *F4,134* = 0.50, *p* = 0.735, *η*^2^ = 0.02; neopterin).

## 4. Discussion

The present study confirmed former results showing higher kynurenine levels and an elevated kynurenine/tryptophan in individuals with BD compared to C as well as higher levels in overweight persons than in normal-weight ones regardless of BD diagnosis. No differences in neopterin levels were found. However, neopterin correlated with kynurenine/tryptophan, which indicates activated cellular immune responses. Furthermore, for the first time, serum concentrations of tryptophan, kynurenine, neopterin, and kynurenine/tryptophan were investigated in a longitudinal setting in BD in a follow-up period with three different time points. Interestingly, the levels did not significantly change over time. Importantly, no illness-specific parameters (euthymia or occurrence of illness episode) had an impact on the course of tryptophan, kynurenine, and kynurenine/tryptophan or neopterin.

The observed positive correlations between neopterin and kynurenine and kynurenine/tryptophan support the assumption of an immune-mediated activation of tryptophan catabolism and thus of IDO-1 activity [20]. However, a minor contribution of other tryptophan catabolizing enzymes cannot be excluded [50]. Under pro-inflammatory, pro-oxidative conditions, tryptophan is more likely converted into kynurenine instead of serotonin, as the latter route requires the presence of the oxidation-labile cofactor tetrahydrobiopterin [51]. Kynurenine is a source of several neuroactive metabolites, including kynurenic acid, 3-hydroxykynurenine, and quinolinic acid. Their exact role in the etiology of psychiatric and neurological disorders is not yet fully explored [52]. BD is associated with low-grade inflammation with elevated pro-inflammatory cytokines and acute-phase proteins [5], supported by the observed differences in IDO-1 activity between BD and C. Likewise, overweight is characterized by inflammation [53].

Tryptophan, kynurenine, and kynurenine/tryptophan levels stayed stable over time in BD patients. Additionally, illness episodes between the study visits and current psychopathology did not impact kynurenine or kynurenine/tryptophan. These findings confirm the hypothesis that low-grade inflammation in BD is a constant phenomenon, occurring not only during manic or depressive episodes but also in euthymia [3]. Therefore, our results indicate that tryptophan, kynurenine, and IDO-1 activity reflected by kynurenine/tryptophan might play a role in pathophysiology in BD but are not necessarily associated with clinical manifestations. However, the medication adoption in illness episodes or the duration of an episode might influence these parameters. In depressed individuals treated in a six-week rehabilitation program, it was previously shown that the kynurenine to tryptophan ratio decreased significantly, however only in the group with a clinical response [54].

Additionally, metabolic and cardiovascular diseases are linked to immune- and inflammatory processes and oxidative stress. These diseases are common comorbidities in BD individuals, primarily explainable with the high occurrence of weight gain resulting in overweight and obesity [8]. Our results show that higher kynurenine to tryptophan ratio as a proxy for IDO-1 activity in overweight participants supports the association between BD, overweight, and inflammation. The kynurenine to tryptophan ratio was the highest in the subgroup of individuals with BD and overweight, followed by individuals with BD and normal weight. In the C group, individuals with overweight also showed higher levels than normal weight. This finding is comparable with the previously reported serum concentration of mentally healthy people [25]. In addition, in a cohort of individuals with BD, tryptophan, kynurenine, and kynurenine/tryptophan were shown to correlate with a craving for carbohydrates [55], which in turn might lead to further weight gain. However, sex [56] and age [57] affect tryptophan metabolism, and the normal weight C sub-group was composed of younger individuals and more females than the other groups. Additionally, oxidative triggers might be involved in the complex mechanisms of BD, inflammation, and comorbidities [58]. ROS are linked to overweight as it leads to lipid peroxidation. Moreover, oxidative stress and decreased antioxidants defense play roles in cardiovascular diseases [59] and metabolic syndrome [60].

Accelerated breakdown of serum tryptophan to kynurenine may lead to reduced circulating tryptophan available for the production of serotonin. According to the monoamine-deficit hypothesis, a shortage of cerebral serotonin is involved in neuropsychiatric symptomatology of depression, mania, and psychosis [15,61]. As the serotonin metabolite melatonin is involved in several processes relevant to BD, such as regulating circadian rhythms, sleep, gut, and immune cell reactivity, a deficit impacts symptomatology and comorbidities [62]. Interestingly, there were negative correlations of kynurenine and kynurenine/tryptophan concentrations and difficulties falling asleep. Further research should elucidate this more by investigating the tryptophan metabolism towards melatonin.

The central catabolites of kynurenine are assumed to affect neurotransmission and neurocognition via neurotoxicity and neuroprotection, respectively. For instance, in microglial cells, kynurenine is converted into 3-hydroxykynurenine by kynurenine-3-monooxygenase and may be further converted to quinolinic acid. Both are primarily considered neurotoxic through oxidative radicals’ release and act as agonists at the N-Methyl-D-Aspartat (NMDA)-receptor. Some tryptophan metabolites, as TDO, have antioxidative activity [63], but others, as 3-hydroxykynurenine, lead to ROS production [64]. The expression and activity of this enzymatic pathway are increased in neuropsychiatric disorders such as schizophrenia and BD and are regulated by pro-inflammatory cytokines [32]. In astrocytes, the kynurenine aminotransferase II converts kynurenine to kynurenic acid which has neuroprotective effects by acting as an NMDA-receptor-antagonist and α7-nicotinic acetylcholine receptor-antagonist. However, an accumulation of kynurenic acid could lead to cognitive disturbances and psychosis [21]. The direction of peripheral kynurenine catabolism towards neurotoxic paths is indicated by decreased kynurenic acid and increased 3-hydroxykynurenine in individuals with BD compared to healthy controls without a mental disorder [65]. A shift towards the hydroxykynurenine arm of the kynurenine pathway was associated with poorer memory performance in a BD group [66].

Some tryptophan catabolites, in particular kynurenine, can activate the aryl hydrocarbon receptor (AhR). This ligand-activated transcription factor activates gene sets involved in various cellular processes such as embryogenesis, transformation, tumorigenesis, and inflammation and is also essential for CNS-relevant processes [62,67]. Metabolic alterations in glucose, fatty acid, and NAD+ metabolism connect AhR signaling and obesity [68]. AhR and its complexes were reported to play a role in acute mania and BD-associated circadian rhythm dysregulations [62].

Many redox-dependent feedback mechanisms are involved in the fine-regulation of the above-mentioned metabolic pathways [7]. Tetrahydrobiopterin, which is produced by most cell types when GTP-CH-I is activated while human monocytes/macrophages produce neopterin, is an essential cofactor for monooxygenases such as tryptophan 5-hydroxylase (TPH) and nitric oxide synthase (NOS), but also of phenylalanine 4-hydroxylase (PAH) and tyrosine 3-hydroxylase (TH). Tetrahydrobiopterin is an oxidation labile. Moreover, cross-pathway regulatory feedback mechanisms that depend on downstream molecules have been identified, e.g., NO inhibits IDO [69], and xanthurenic acid, a metabolite formed from 3-hydroxykynurenine, has been shown to inhibit sepiapterin reductase, the final enzyme in de novo tetrahydrobiopterin synthesis [70]. The relevance of this complex crosstalk needs to be further evaluated in the psychiatric context. This analysis did not confirm former results that neopterin is elevated in affective disorders. Nonetheless, the correlation of neopterin with tryptophan catabolites indicates the activation of inflammatory pathways. Additionally, it can be assumed that inflammatory processes in the context of somatic disorders as cardiovascular diseases play a significant role [15]. Therefore, further studies with a greater number of participants with and without comorbidities are needed.

Summarizing, this study proved the presence of increased tryptophan catabolism along the kynurenine axis in BD, most likely due to immune-mediated activation of IDO-1 as the increased kynurenine to tryptophan ratio correlated with elevated neopterin concentrations. We hypothesize that this indicates a shift in the tryptophan metabolism from serotonin to the kynurenine pathway and that somatic comorbidities facilitate such metabolic dysregulation or vice versa. Overweight and metabolic and cardiovascular disease, which are common comorbidities of BD, may also trigger inflammation. Therefore, the normalization of tryptophan metabolism might positively impact symptomatology, cognition, and comorbidities.

### Limitations

There are several limitations of this study. First, psychopharmaceuticals might have an impact on tryptophan and kynurenine. Nevertheless, controlling for every medication (classes, substances, doses, and duration) would have yielded a myriad of permutations of treatment combinations not fit for statistical analysis. At least, all individuals with BD in this study were treated with psychopharmacology and psychotherapy according to clinical guidelines. Moreover, although chronic inflammatory diseases were an exclusion criterion, the included individuals suffered from heterogeneous somatic comorbidities and had differing pharmaceutical requirements, which might have influenced inflammatory and clinical parameters as well as tryptophan metabolism. Although we aimed to investigate a matched C group, there were differences between BD and C in age and BMI. Older age was shown to be associated with increased tryptophan metabolism [71]. Metabolites were measured in peripheral blood and therefore cannot draw conclusions about central processes. However, peripheral measurements may be seen as a proxy for monitoring disease activity and treatment response. The other tryptophan metabolism arm towards serotonin and melatonin and further catabolites of kynurenine such as 3-hydroxykynurenine and quinolinic acid were not investigated. Serotonin levels were not measured, and therefore the shift to the kynurenine pathway can only be assumed.

## 5. Conclusions

The present study investigated the tryptophan metabolism in BD, displaying higher kynurenine and kynurenine/tryptophan as a proxy for IDO-1 activity than in C and higher levels in overweight persons than in normal weight individuals. As both the increased IDO-1 activity and the shift in the tryptophan metabolism from serotonin to the kynurenine pathway in BD is associated with weight, decreases of serotonin and melatonin may present a risk for neurotoxicity. Therefore, interventions to reduce the inflammatory background and thus upstream activator of the tryptophan kynurenine axis may normalize metabolite levels and beneficially influence symptomatology, cognition and somatic comorbidities. In our study, no large illness-specific parameters such as euthymia or occurrence of illness episode impacted on the course of tryptophan, kynurenine, and kynurenine to tryptophan ratio. However, more research with large sample sizes in longitudinal settings, including psychopharmaceutical treatment, is highly recommended.

## Figures and Tables

**Figure 1 antioxidants-10-01795-f001:**
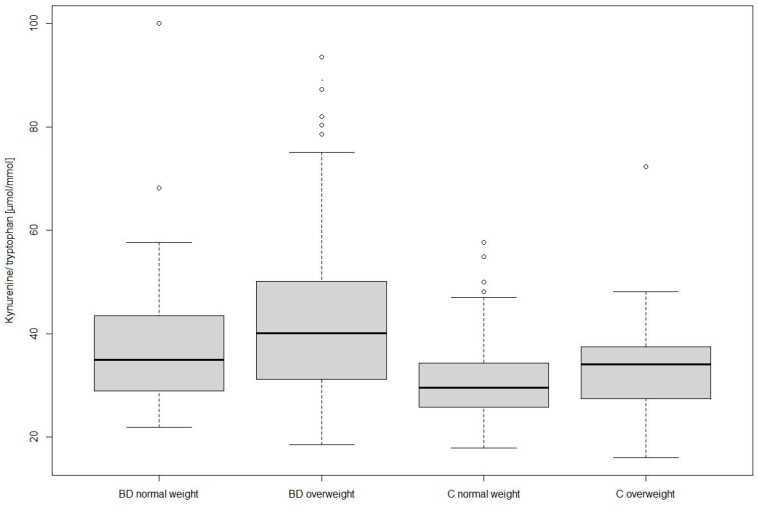
Kynurenine to tryptophan ratio in individuals with bipolar disorder (BD) and controls (C) with normal and overweight.

**Table 1 antioxidants-10-01795-t001:** Demographic data, tryptophan concentrations, its metabolite kynurenine, and neopterin stratified by weight in BD patients and controls.

	BD		C		Statistics		
	Normal Weight (*n* = 80) (*M* ± *SD*)	Overweight (*n* = 146) (*M* ± *SD*)	Normal Weight (*n* = 97) (*M* ± *SD*)	Overweight (*n* = 45) (*M* ± *SD*)	*F_group_*	*F_weightgroup_*	*F_group × weight group_*
Age [years]	41.33 (13.21)	45.36 (12.73)	33.31(14.35)	41.69 (14.10)	**14.57 ****	**16.38 ****	2.02
Sex	40% ♂ 60% ♀	58.2% ♂ 41.8% ♀	33.0% ♂ 67.0% ♀	46.7% ♂ 53.3% ♀			
BMI [kg/m^2^]	22.61 (1.69)	30.54 (5.12)	21.68 (1.77)	29.14 (3.43)	**7.82 ****	**341.63 ****	0.32
Tryptophan [µmol/L]	55.69 (10.76)	57.74 (10.43)	62.54 (10.55)	61.40 (7.80)	^#^ **18.12 ****	^#^ 0.55	^#^ 0.79
Kynurenine [µmol/L]	2.04 (0.62)	2.43 (0.89)	1.90 (0.46)	2.04 (0.51)	^#^ 5.39	^#^ **4.15 ***	^#^ 3.17
Kynurenine/tryptophan [µmol/mmol]	37.22 (11.68)	42.39 (14.23)	30.86 (7.95)	33.73 (9.40)	^#^ **19.17 ****	^#^ 2.26	^#^ 2.26
Neopterin [nmol/L]	^a^ 6.38 (4.59)	^b^ 6.87 (3.17)	^c^ 5.93 (1.71)	^d^ 6.09 (1.67)	0.10	0.11	0.19

Note: BD = bipolar disorder; C = controls; BMI = body mass index; ^a^ *n* = 78; ^b^
*n* = 139; ^c^ *n* = 97; ^d^ *n* = 29; Statistically significant group differences by (M)ANOVAs in bold letters; * = *p* < 0.05; ** = *p* < 0.01; ^#^ = MANCOVA with age and sex as covariates.

**Table 2 antioxidants-10-01795-t002:** Demographic data of individuals with BD (*n* = 75) in a longitudinal setting of three measurement time points.

	Mean t1–t3	t1	t1–t2	t2	t2–t3	t3
Age [years] (*M* ± *SD*)	47.16 (13.29)					
Sex	53.3% male 46.7% female					
BMI [kg/m^2^] (*M* ± *SD*)	28.63 (5.35)	28.31 (5.30)		28.95 (5.50)		28.90 (5.34)
Weight group	25.3% normal 74.7% overweight					
Time [days] (*M* ± *SD*)			317.51 (176.27)		384.15 (227.52)	
Euthymic (HAMD < 14 + YMRS < 9) (YES)		74.7%		92.0%		84.0%
Illness episode None Mania Depression			58.7% 12.0% 29.3%		53.3% 16.0% 29.3%	
Tryptophan [µmol/L] (*M* ± *SD*)		58.44 (9.46)		57.89 (9.99)		59.39 (9.32)
Kynurenine [µmol/L] (*M* ± *SD*)		2.23 (0.67)		2.15 (0.65)		2.21 (0.65)
Kynurenine/tryptophan [µmol/mmol] (*M* ± *SD*)		38.34 (10.83)		37.72 (11.90)		37.73 (12.04)
Neopterin ^a^ [nmol/L] (*M* ± *SD*)		7.09 (3.33)		7.16 (3.39)		7.48 (4.31)

Note: ^a^ *n* = 68; BD = bipolar disorder; BMI = body mass index; HAMD = Hamilton Depression Scale; YMRS = Young Mania Rating Scale.

## Data Availability

The data presented in this study are available on request from the corresponding author, due to ongoing analysis.

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
