# Peer review of "Tryptophan Metabolism in Bipolar Disorder in a Longitudinal Setting"

_antioxidants, 2021, doi:10.3390/antiox10111795_

Round 1
Reviewer 1 Report
Overview and general recommendation:
The present study revealed higher kynurenine and kynurenine/tryptophan as a proxy for IDO-1 activity than in C and higher levels in overweight persons than normal-weight individuals. Furthermore, for the first time, serum concentration of tryptophan, kynurenine, neopterin, and kynurenine/tryptophan did not significantly change over time. This well-written paper contains interesting and innovative results indirectly relevant to oxidant stress, which merits publication in this journal. For the benefit of readers, however, a few points need clarifying, and certain statements require further justification.
Major concern
Metabolite of tryptophan and kynurenine has a crucial role in circadian rhythms, as the authors discussed. More detail should be added associations between insomnia evaluated by HAMD and serum concentration of tryptophan, kynurenine, neopterin, and kynurenine/tryptophan.
Minor concern
It should be cautiously considered that the authors evaluated not IDO-1 but kynurenine to tryptophan ratio as a proxy for IDO-1 activity. For example, “Summarizing, the increased IDO-1 activity in BD was confirmed in this study” was very confusing in line 1 of the 8th paragraph of the Discussion section.
Author Response
Thank you very much for your time, your friendly comments and your valuable input.
Major concern
Metabolite of tryptophan and kynurenine has a crucial role in circadian rhythms, as the authors discussed. More detail should be added associations between insomnia evaluated by HAMD and serum concentration of tryptophan, kynurenine, neopterin, and kynurenine/tryptophan.
That is indeed an interesting and important topic. Therefore, we conducted correlation analyses with the HAMD items concerning sleep and added the following:
Methods/materials: The items 4-6 (each 0-2 points) measure insomnia, in particular difficulties falling asleep, being disturbed during the night and wakes in early morning hours.
Methods/statistics: For the individuals with BD at t1 only, partial correlations with tryptophan, kynurenine, ratio, neopterin and the HAMD items 4-6 witch the covariates age, sex, BMI were conducted.
Results: Of 204 individuals with BD at t1 the mean for falling asleep problems was 0.45 (± 0.68), for sleep disruptions during night 0.41 (± 0.66) and for wakes in the morning hours 0.30 (± 0.63). No correlations between tryptophan or neopterin with the insomnia items of the HAMD were found. There was a significant negative correlation between the HAMD item 4 for difficulties falling asleep and kynurenine (r = -.150, p = .033). as well as kynurenine/tryptophan (r = -.170, p = .016). The other items 5 and 6 did not correlate significantly.
Discussion: Interestingly, there were negative correlations of kynurenine as well as kynurenine/tryptophan concentrations and difficulties falling asleep. Further research should elucidate this more by investigating also the tryptophan metabolism towards melatonin.
Limitations: The other tryptophan metabolism arm towards serotonin and melatonin as well as further catabolites of kynurenine such as 3-hydroxykynurenine and quinolinic acid were not investigated.
Minor concern
It should be cautiously considered that the authors evaluated not IDO-1 but kynurenine to tryptophan ratio as a proxy for IDO-1 activity. For example, “Summarizing, the increased IDO-1 activity in BD was confirmed in this study” was very confusing in line 1 of the 8th paragraph of the Discussion section.
Thank you for that advice. We added the following to the discussion: The observed positive correlations between neopterin and kynurenine as well as kynurenine/tryptophan support the assumption of an immune-mediated activation of tryptophan catabolism and thus of IDO-1 activity [20]. However, a minor contribution of other tryptophan catabolizing enzymes cannot be excluded [Badawy, A.A. Kynurenine pathway and human systems. Exp Gerontol. 2020, 129:110770. doi: 10.1016/j.exger.2019.110770.]
Additionally, we revised the sentence you suggested: Summarizing, this study proofed the presence of increased tryptophan catabolism along the kynurenine axis in BD, most likely due to immune-mediated activation of IDO-1 as the increased kynurenine to tryptophan ratio correlated with elevated neopterin concentrations.
Furthermore, we revised the following the 3rd paragraph of the discussion: Therefore, our results indicate that tryptophan, kynurenine, and IDO-1 activity reflected by kynurenine/tryptophan might play a role in pathophysiology in BD but not necessarily associated with clinical manifestations. Additionally, the sentence in the 4th paragraph was revised: Our results showing a higher kynurenine to tryptophan ratio as a proxy for IDO-1 activity …
Reviewer 2 Report
This is an original paper in which, authors assess tryptophan metabolism in BD compared to controls, stratified by weight classes, in a longitudinal setting, dependent on the incidence of BD episodes.
In Introduction
Authors only described the part of the kynurenine pathway tryptophan-ido-kynurenine, not enough about kynurenic acid or 3-hydroxykynurenine, which have a very important part in depression. Why is it omitted? more see: Myint et al., 2007; Erhardt et al., 2017)
In Laboratory methods
no validation parameters: linearity, the limit of detection (LOD), LOQ, precision, and recovery are missing. Complete, please.
In Discussion
line 298 "While kynurenine itself is inactive" in my opinion kynurenine is a very active endogenous substance (see e.g.: Kolodziej et al. 2011; Opitz et al., 2011). Additionally, the authors should expand the discussion to include AhR receptor which is involved in multiple physiological and pathophysiological functions: energy metabolism - obesity (see: Bock, 2019).
Author Response
Dear editors and reviewer,
Thank you for your friendly comments concerning our paper “Tryptophan metabolism in bipolar disorder in a longitudinal setting”.
We revised the paper based on all reviewers’ recommendations. We hope our changes meet your expectations and that the manuscript is now ready to be published in Antioxidants-special issue: Tryptophan metabolism in health and disease.
Explanations: Sentences that localize changes in the manuscript are underlined in the comments. Changes are marked using the track-changes function of MS Word in the “marked” manuscript-file.
Dear reviewer, thank you very much for your time, your comments and your valuable input.
In Introduction: Authors only described the part of the kynurenine pathway tryptophan-ido-kynurenine, not enough about kynurenic acid or 3-hydroxykynurenine, which have a very important part in depression. Why is it omitted? more see: Myint et al., 2007; Erhardt et al., 2017)
This is in indeed a very important topic for further investigations. The aim of these analyses was first to replicate the tryptophan baseline levels in BD compared to controls and investigate them in a longitudinal setting. Second the focus is to determine the activation of the kynurenine pathway in its first step to demonstrate a base for further down step metabolisms investigations. Importantly, even if it seems likely that there are relations between peripheral tryptophan catabolites and CNS levels current research focus on animal models and human cerebrospinal fluid data is still rare. Data from animal studies showed differences of kynurenine catabolites in different brain areas (for example Erhardt et al., 2017). Additionally, differences in peripheral and central kynurenic acid levels in BD individuals were found (Sellgren et al., 2019). Therefore, the fully insight about processes in the CNS of further tryptophan catabolites cannot be presented with peripheral samples. Nonetheless, we revised the following in the manuscript:
Introduction (line 95): The same study group found increased kynurenic acid only in CSF and not in peripheral plasma in individuals with BD and previous psychosis compared to controls, while the CSF kynurenic acid concentrations where unchanged in BD patients with ongoing depression [Sellgren et al., 2019]. … Moreover, the availability of CSF is usually limited.
Kynurenine is further metabolized into several neuroactive compounds as of most significance kynurenine acid, 3-hydroxykynurenine and quinolinic acid, which may impact neuropsychiatric disorders as BD [Myint, 2012]. Importantly, psychosocial stress does also affect tryptophan metabolism [Michels et al., 2018; o’Farrell & Hakin, 2017; Myint et al., 2007].
Discussion: … However, a minor contribution of other tryptophan catabolizing enzymes cannot be excluded [50]. Under pro-inflammatory, pro-oxidative conditions tryptophan is more likely converted into kynurenine instead of serotonin, as the latter route requires the presence of the oxidation-labile cofactor tetrahydrobiopterin [51]. Kynurenine is a source of several neuroactive metabolites, including kynurenic acid, 3-hydroxykynurenine and quinolinic acid. Their exact role in the etiology of psychiatric and neurological disorder is not yet fully explored [52].
In Laboratory methods: no validation parameters: linearity, the limit of detection (LOD), LOQ, precision, and recovery are missing. Complete, please.
The detection limits and further references were added: Limits of detection were 0.1 µmol/l for tryptophan and 0.5 µmol/l for kynurenine. Further information on method validation was reported [Laich et al., 2002; Widner et al., 1999].
In Discussion: line 298 "While kynurenine itself is inactive" in my opinion kynurenine is a very active endogenous substance (see e.g.: Kolodziej et al. 2011; Opitz et al., 2011).
Thank you for that advise. We revised the sentence: The central catabolites of kynurenine are assumed to affect neurotransmission and neurocognition via neurotoxicity and neuroprotection, respectively.
Additionally, the authors should expand the discussion to include AhR receptor which is involved in multiple physiological and pathophysiological functions: energy metabolism - obesity (see: Bock, 2019).
Thank you for your valuable input. We added the following to the discussion: Some tryptophan catabolites, in particular kynurenine can activate the aryl hydrocarbon receptor (AhR), a ligand-activated transcription factor that activates genesets involved in a variety of cellular processes such as embryogenesis, transformation, tumorigenesis and inflammation, and is also important for CNS-relevant processes [Anderson et al., 2016; Opitz et al., 2011]. Metabolic alterations in glucose, fatty acid and NAD+ metabolism connect AhR signaling and obesity [Bock, 2019]. AhR and its complexes were reported to play a role in acute mania and BD associated circadian rhythm dysregulations [Anderson et al., 2016].
Round 2
Reviewer 1 Report
I recognized that the authors have responded to all my comments appropriately. The manuscripts are improved.
Author Response
Thank you very much for appreciating our revisions.
Reviewer 2 Report
I know the validation methods, but in my opinion, the results of validation parameters should be covered in the Method paragraph.
Author Response
Thank you for your additional input. According to your suggestions we added the following to the methods: "Optimal chromatographic separation was achieved on a LiChroCART RP-18 endcapped column (55-4, 3 µm, Merck, Germany) by using 15 mmol/L acetic acid-sodium acetate solution (pH = 4.0) as mobile phase. Validation parameters were linearity (R2 ≥ 0.998 for both tryptophan and kynurenine), interday precision (≤ 1,8% for tryptophan and ≤ 4.2% for kynurenine) and recovery (99.3% ± 0.4% (mean SD) for tryptophan and 102.3% ± 1.0% for kynurenine) [45]."